# Mind the Exit Pupil Gap: Revisiting the Intrinsics of a Standard Plenoptic Camera

**DOI:** 10.3390/s24082522

**Published:** 2024-04-15

**Authors:** Tim Michels, Daniel Mäckelmann, Reinhard Koch

**Affiliations:** Department of Computer Science, Kiel University, 24118 Kiel, Germanyrk@informatik.uni-kiel.de (R.K.)

**Keywords:** plenoptic cameras, light field, calibration, refocusing, simulation

## Abstract

Among the common applications of plenoptic cameras are depth reconstruction and post-shot refocusing. These require a calibration relating the camera-side light field to that of the scene. Numerous methods with this goal have been developed based on thin lens models for the plenoptic camera’s main lens and microlenses. Our work addresses the often-overlooked role of the main lens exit pupil in these models, specifically in the decoding process of standard plenoptic camera (SPC) images. We formally deduce the connection between the refocusing distance and the resampling parameter for the decoded light field and provide an analysis of the errors that arise when the exit pupil is not considered. In addition, previous work is revisited with respect to the exit pupil’s role, and all theoretical results are validated through a ray tracing-based simulation. With the public release of the evaluated SPC designs alongside our simulation and experimental data, we aim to contribute to a more accurate and nuanced understanding of plenoptic camera optics.

## 1. Introduction

Plenoptic cameras as initially described by Lippmann [1] and Ives [2] combine a traditional camera with an additional microlens array (MLA) located between the main lens and the sensor. Over the years, two primary designs have been extensively studied and brought to market, the standard plenoptic camera (SPC) [3,4] and the focused plenoptic camera (FPC) [5,6], which mainly differ in the microlens focus distance. Due to their earlier commercialization, larger angular resolution, and simpler decoding process, SPCs remain popular despite certain disadvantages in terms of spatial resolution and depth of field when compared to the multi-focus variant of FPCs [6]. Classical applications for SPCs include depth reconstruction [3] and post-capture refocusing from single shots [4]; as a first step to achieve these, the raw 2D image of a plenoptic camera is usually de-multiplexed and resampled into a 4D light field [7], as shown in Figure 1. For this reparametrization procedure, knowledge about the exact position of each microlens image center (MIC) is crucial, as any inaccuracies in their locations can result in computational errors affecting the quality of the refocused images [8]. Furthermore, a formal connection between the MICs and the plenoptic camera optical setup is required to relate the light field within the camera to the optical reality outside the camera, e.g., to find the correct refocusing parameters for the desired object distance [7,8].

Over the past two decades, a number of studies have delved into the topic of processing plenoptic camera images; however, these have often considered the MICs to be determined by the main lens center or its principal planes as a consequence of reducing the main lens to a simple thin lens. This assumption oversimplifies the actual optics involved. A more accurate representation acknowledges the role of the exit pupil in determining these image centers, as observed in studies by Hahne et al. [8,9]. Despite these advancements, the exit pupil is often ignored in studies relating the light field within the camera to the 3D scene in front of the camera.

**Figure 1 sensors-24-02522-f001:**
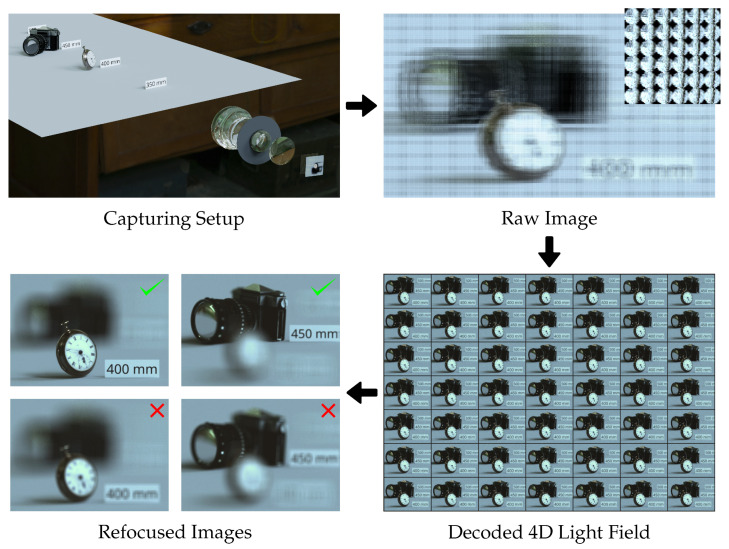
Exemplary pipeline for SPC post-shot refocusing. A scene is captured by a virtual SPC, shown here without the housing. The resulting raw image consists of a large number of microlens images and is subsequently decoded into a 4D light field representation, which can be visualized by a subset of the sub-aperture images [7]. By resampling the light field, a refocused image can be created [4]. The correctly focused images shown here were created based on parameters considering the exit pupil as described in Section 2, while the slightly defocused image are results from the directly calculated parameters based on [10] without any exit pupil consideration.

In this context, our work aims to again highlight the importance of the exit pupil. To this end, a paraxial model of the SPC under consideration of the exit pupil is first described, directly relating the refocusing shift [4] to the object distance. The expected errors of models ignoring the exit pupil are formally analyzed and later verified through a ray tracing-based simulation of various plenoptic cameras in Blender [11] using real lens data. Subsequently, multiple works in the domain of plenoptic camera calibration are revisited and examined with respect to the need for a more complex lens model. More specifically, the popular work of Dansereau et al. [7] is first revisited, along with the works of Zhang et al. [12] and Monteiro et al. [13] building upon Dansereau’s ideas. In these cases, it can be concluded that the parameters of the respective calibration models are sufficiently general to permit the simplicity of a main lens model without considering the exit pupil. However, this only holds true because these works do not require a specific interpretation of the model parameters. On the other hand, for the work of Pertuz et al. [10], which also employs the decoding from [7] for metric distance measurement, it is shown that the oversimplified main lens model leads to an incorrect interpretation of the metric refocusing model parameters. In summary, our contributions are:A formal deduction of the connection between object distance and sub-aperture image shift considering the exit pupil.A model for the errors resulting from ignoring the exit pupil in this relation.An analysis of the exit pupil’s role in popular works on SPC calibration [7,10,12,13,14].Publicly available SPC designs [15] and a camera simulation framework [16] based on Blender [11] supporting a large database of lens designs and enabling quick generation of new plenoptic camera setups.

### 1.1. Related Work

**Plenoptic Cameras:** There are two primary design concepts of plenoptic cameras which have been extensively studied and brought to market. The first, known as the standard plenoptic camera (SPC), was described by Adelson and Wang [3] and later commercialized by Ng [4]. It requires the microlenses to be focused at infinity, i.e., the MLA-to-sensor separation must match the microlens focal length. Consequently, for a scene object placed at the SPC’s focus distance, all sensor pixels behind a single microlens effectively capture a nearly identical segment of this object, albeit from slightly varied perspectives. This results in large angular resolution but low spatial resolution.

Later, the focused plenoptic camera (FPC) was presented by Lumsdaine and Georgiev [17] and extended by Perwass and Wietzke [6] to feature multifocal MLAs for extended depth of field. For an FPC, instead of directly dissecting the scene into its directional components, the microlenses are focused at the scene’s virtual image inside the camera. This arrangement is advantageous in maintaining a greater portion of the conventional camera’s spatial resolution, albeit at the expense of a decrease in angular resolution compared to SPCs.

Despite the advantages of FPCs, especially when combined with multifocal MLAs, in this work we chose to focus on SPCs due to their straightforward image processing pipeline. More specifically, in order to show the effect of the exit pupil, the application of post-shot refocusing is used throughout this work. While this is a possible application for an FPC, the process involves direct [18] or indirect [19] depth estimation, which can introduce artifacts that affect subsequent processing steps. In contrast, for a well-configured SPC the post-shot refocusing only comprises a demultiplexing step [7] for the raw image and a subsequent shift-and-sum procedure for the resulting sub-aperture images. The simplicity of this pipeline reduces the quantity of artifacts resulting from complex interpolation and optimization steps, allowing for more direct analysis of the exit pupil’s effects. It should be noted, however, that this work is intended to be a starting point for further research which will analyze and improve FPC calibration algorithms suffering from the same model flaws as their SPC pendants.

**SPC Calibration:** The calibration of plenoptic cameras plays a crucial role in relating the captured light field within the camera to the 3D world in front of the camera. To this end, Dansereau et al. [7] presented a method for demultiplexing the raw image of an SPC into a 2D array of sub-aperture images and using these for the geometric calibration. Due to its popularity and accessibility in the form of a Matlab toolbox, this work is still being used as a base for publications concerning the processing of SPC data. With respect to calibration models, Zhang et al. [12] and Monteiro et al. [13] modified the ideas of [7] to associate a plenoptic camera with an equivalent multi-camera array. Both of these works made direct use of the decoding process proposed by Dansereau et al. [7]. Pertuz et al. [10] followed the same approach in proposing a focus-based metric for depth estimation.

Due to the popularity of the demultiplexing process from [7], the revision of the previous literature in Section 4 focuses on this method and the approaches based on it. Nevertheless, there has been further work, which in part also reduces the main lens to a thin lens, ignoring the effects of the exit pupil. One such calibration approach, which directly uses the line features in the microlens images, is the one presented by Bok et al. [20]. Using a similar model, Zhao et al. [21] performed SPC calibration based on the plenoptic disc features. Both Thomasen et al. [22] and Suliga and Wrona [23] directly estimated the MLA pose and microlens pitch; however, neither related the captured light field to the scene-side light field. Similar to the approaches related to [7], these works assume the microlens images centers to be projections of the main lens center, i.e., the center of the camera-side principal plane. On the other hand, Hahne et al. [8,9] described the refocusing distance based on known main lens and MLA parameters under consideration of the exit pupil. Further improvements to aspects of the calibration pipeline which acknowledge the exit pupil have been presented by Schambach et al. [24], who increased the MIC detection accuracy, and Mignard-Debise and Ihrke [25], who analyzed the effect of vignetting on calibration models.

Of these works, Hahne et al. [9] considered the exit pupil and its connection to the microlens image geometry in a similar fashion to this work; however, they did not situate this within the direct context of pre-existing calibration methods by a comparative evaluation, nor provide an analysis of the expected errors resulting from oversimplified lens models. In order to further validate our model, which establishes the refocusing distance in terms of the two-plane parameterization, its equivalence to the chief ray intersection model in [9] is formally proven in Section 4.1.

**FPC Calibration:** Despite this work focusing on SPCs, as explained above, there is relevant work to be found in the related field of FPC calibration. Johannsen et al. [26] described a metric reprojection model for FPCs incorporating a radial distortion model. This was further enhanced by Heinze et al. [27], who included the tilt and shift of the main lens as well as multi-focus MLAs. Further improvements to the distortion model were presented by Zeller et al. [28]. All of these approaches are based on the reconstruction of the virtual scene between the MLA and the main lens followed by association of these virtual 3D points with known scene points. In contrast, Noury et al. [29] proposed an approach that works directly on the microlens images, i.e., by associating the scene points with their projections on the sensor without the intermediate step of calculating virtual depths. This method, however, is limited to single-focus FPCs. and models the microlenses as simple pinholes. Nousias et al. [30], on the other hand, featured a more complete microlens model in their work and directly included the estimation of multiple microlens focal lengths in their approach. Wang et al. [31] presented a two-step model consisting of a forward projection from the scene into the camera and a second projection from the virtual image to the sensor. More recently, Labussiere et al. [32] proposed simultaneous calibration of the different microlens types in a multi-focus plenoptic camera by incorporating defocus blur into the features used for parameter optimization.

None of the listed methods for FPC calibration directly consider the exit pupil, and while most of these works, all of which require the identification of MICs, incorporate a scaling between the grid of microlens centers and the grid of MICs, this is usually a result of projecting the main lens center, i.e., the center of the camera-side principal plane, through the microlens centers. However, as observed by Hahne et al. [8,9] for SPCs and confirmed in Section 5, the MICs actually result from a projection of the exit pupil’s center. Thus, using the distance between the simplified main lens plane and the MLA for both the image formation model and the calculation of MICs could inadvertently reduce the degrees of freedom of the model. While this might be desirable in terms of increased stability during parameter optimization, such reduction should be analyzed for FPC models. For reasons of clarity and comprehensibility, we decided against including the topic of FPC calibration in this work, and leave this for future efforts.

**Lens Models and Simulation:** In the domain of ray tracing-based camera simulation, realistic main lens models which consider all lens components and their respective properties have been used for over two decades, either explicitly by direct modeling, as in Kolb et al. [33] and Wu et al. [34], or implicitly via learned black-box lens models, as proposed in Zheng et al. [35]. Regarding plenoptic cameras, most previous works have used oversimplified models for rendering, such as pinhole cameras or multi-camera arrays modeling the MLA without a model for the main lens [36,37,38]. More recently, Nürnberg et al. [39] as well as our own group [40] have provided simulations of plenoptic cameras without oversimplifying the main lens. Due to this familiarity, we extended our previous work for our synthetic experiments.

### 1.2. Organization

In Section 2, the general lens model and two-plane parameterization are explained first, before deducing the refocusing model under consideration of the exit pupil. Subsequently, Section 3 provides a formal analysis of the expected errors when dismissing the exit pupil. In Section 4, previous works are revisited with a focus on the need for more complex lens models. Finally, our deductions are validated with synthetic experiments in Section 5.

## 2. SPC Optics

### 2.1. Preliminaries—Lens Models

The thin lens model describes a lens by assuming it as infinitely thin and only refracting light at a single lens plane. The relation between the real scene and the lens image in this model is described by the equation
(1)$$\frac{1}{f_{M}}=\frac{1}{o}+\frac{1}{i}\;,$$
where $f_{M}$ is the focal length of the lens, o is the object distance, and i is the image distance, with both measured from the refraction plane. This concept can be extended to a thick lens model by expanding the refraction plane into two principal planes, $H_{\mathrm{scene}}$ and $H_{\mathrm{cam}}$, between which a traced light ray is considered to run parallel to the optical axis [41,42]. Furthermore, a combination of thick lenses, such as the main lens of a plenoptic camera, can again be represented as a single thick lens [42]. As visualized in Figure 2, the object distances o and image distance i are then measured based on the positions of the principal planes, and Equation (Equation 1) remains valid.

In addition to this model, it is possible to consider the exit pupil, i.e., the image of the aperture stop viewed towards the image plane. It defines the size and location of the virtual aperture in the optical system [42] and, as pointed out by Hahne et al. [8,9], determines the positions of the microlens image centers (MIC) on the sensor. As empirically shown in Section 3, the exit pupil and $H_{\mathrm{cam}}$ rarely coincide; accordingly, a systematic error can be introduced when a plenoptic camera image is de-multiplexed based on MICs that are incorrectly estimated under the premise that the main lens follows the thin lens model without considering the exit pupil.

### 2.2. Preliminaries—Light Field Parametrization

Despite being a standard tool when working with light field data, the two-plane parameterization as described by Levoy and Hanrahan [43] and used in various popular works, including the work of Dansereau et al. [7] and Ng et al. [4] is reiterated in this section for two reasons. First, the previous descriptions do not consider the exit pupil; second, the literature is not consistent in terms of the underlying data representation. While [7] used the raw camera image indexed by integer pixel coordinates as the basis for their description, [4] assumed known metric coordinates for every pixel. We follow the approach of [7] in order to facilitate the reproduction of our results.

For a given SPC following the thick lens model with an exit pupil as visualized in Figure 2, the light field inside this camera can be parameterized using two planes: the MLA, which serves as virtual sensor plane, and the exit pupil plane, which can be interpreted as a virtual lens plane. By following the decoding process of Dansereau et al. [7], the 4D light field can be parameterized as $L_{F}\left(i,j,k,l\right)$, with integer indices (k,l) for the uniformly sampled sub-aperture image and (i,j) for the pixel coordinates in that image.

The corresponding metric parameterization ${\tilde{L}}_{F}\left(s,t,u,v\right)$ describes the intensity of light captured at the MLA plane point (s,t,d) coming from the exit plane point (u,v,X). In accordance with Ng et al. [4,44], the distance between these two parameterization planes is denoted as F:=d−X. Note, however, that Ng et al. [4] implicitly assume X=0, as the main lens in their work is modeled by a thin lens. While this is approximately correct for the two tested main lenses in [4], a Zeiss Planar T* 2/80 with $X=7.8\;\mathrm{mm}=0.098\cdot f_{M}$ and a Zeiss Sonnar T* 2.8/140 with $X=-4\;\mathrm{mm}=-0.029\cdot f_{M}$ (compare [45,46]). Section 3 shows that this assumption does not hold in general.

To calculate the metric parameterization from a given integer parameterization, note that the pixel pitch ${∆}_{ST}$ of the virtual sensor, i.e., the step size in the ST-plane, corresponds to the microlens pitch, i.e., ${∆}_{ST}=d_{\mathrm{ML}}$; as shown in Figure 3, the step size in the virtual lens plane, i.e., the UV-plane, can be calculated by means of the triangle equality as ${∆}_{UV}=\frac{s_{px}\cdot F}{f_{m}}$, where $s_{px}$ and $f_{m}$ respectively denote the pixel size and the microlens focal length. With these step sizes, the light field parameterized in metric coordinates (s,t,u,v) is provided by
(2)$${\tilde{L}}_{F}\left(s,t,u,v\right)=L_{F}\left(\frac{s}{{∆}_{ST}},\frac{t}{{∆}_{ST}},\frac{u}{{∆}_{UV}},\frac{v}{{∆}_{UV}}\right).$$
Note that the metric coordinates (s,t,u,v) might not be integer multiples of their respective step sizes; accordingly, querying the corresponding values from the integer parameterization $L_{F}$ could require additional interpolation steps.

With the described light field parameterization, we can now reproduce the resampling steps necessary to refocus the image by moving the virtual sensor plane while considering the position of the exit pupil.

### 2.3. Light Field Refocusing with Exit Pupil

In order to refocus the virtual sensor image onto an object at a distance o measured from $H_{\mathrm{scene}}$, the virtual sensor needs to be placed at a distance i measured from $H_{\mathrm{cam}}$ according to the thin lens Equation (Equation 1). This corresponds to a distance $F^{\prime }:=i-X$ between the UV-plane (exit pupil) and the virtual sensor, as visualized in Figure 4. By defining the refocusing parameter $\alpha =\frac{F^{\prime }}{F}$ as in [4], the thin lens equation can be applied to deduce
(3)$$\alpha =\frac{F^{\prime }}{F}=\frac{\frac{o\cdot f_{M}}{o-f_{M}}-X}{d-X}=\frac{o\cdot \left(f_{M}-X\right)+f_{M}\cdot X}{\left(o-f_{M}\right)\left(d-X\right)}.$$

For a given integer 4D light field $L_{F}\left(i,j,k,l\right)$ based on sub-aperture images, as in [7], the relationship between the virtual sensor movement specified by α and the resulting disparity at the original ST-plane is described by the following. While the general deduction is similar to [4], for reproducibility the following calculations are based on integer indexing.

As shown in Figure 4, the metric light field value ${\tilde{L}}_{F^{\prime }}\left(s^{\prime },t^{\prime },u,v\right)$ for the modified sensor plane placed at a distance $F^{\prime }$ from the exit pupil can be calculated as follows:(4)$$\begin{array}{cc} \; & {\tilde{L}}_{F^{\prime }}\left(s^{\prime },t^{\prime },u,v\right)={\tilde{L}}_{F}\left(u+\frac{s^{\prime }-u}{\alpha },v+\frac{t^{\prime }-v}{\alpha },u,v\right)\\ \; & ={\tilde{L}}_{F}\left(u\left(\frac{\alpha -1}{\alpha }\right)+\frac{s^{\prime }}{\alpha },v\left(\frac{\alpha -1}{\alpha }\right)+\frac{t^{\prime }}{\alpha },u,v\right). \end{array}$$
Ignoring the image magnification introduced by the movement of the virtual sensor, i.e., setting the step size for the $S^{\prime }T^{\prime }$-plane to $\alpha \cdot {∆}_{ST}$ and defining $∆:=\frac{{∆}_{UV}}{{∆}_{ST}}$, it can be deduced that the integer parameterization $L_{F^{\prime }}\left(i,j,k,l\right)$ for the modified sensor plane corresponds to
$$L_{F}\left(k\cdot ∆\left(1-\frac{1}{\alpha }\right)+i,l\cdot ∆\left(1-\frac{1}{\alpha }\right)+j,k,l\right).$$
At this point, the pixel shift *S* between neighboring sub-aperture images required to refocus onto the desired distance o can be calculated for a given value α as follows:(5)$$S\left(\alpha \right)=∆\cdot \left(1-\frac{1}{\alpha }\right).$$
Plugging Equation (Equation 3) into Equation (Equation 5) then yields the direct relation between the object distance o and the disparity *S* as
(6)$$\begin{array}{c} S\left(o\right)=∆\cdot \frac{o\left(f_{M}-d\right)+f_{M}\cdot d}{o\left(f_{M}-X\right)+f_{M}\cdot X}. \end{array}$$
This model can be easily reverted to calculate the object or refocusing distance based on a given sub-aperture image shift via
(7)$$\begin{array}{c} o\left(S\right)=\frac{f_{M}\cdot \left(d\cdot ∆-S\cdot X\right)}{S\cdot \left(f_{M}-X\right)-∆\left(f_{M}-d\right)}. \end{array}$$

## 3. Error Analysis

In the following, we analyze the error that can be anticipated when ignoring the exit pupil, i.e., when setting X=0. First, we define the scaling between the ST and UV planes in under this assumption as
(8)$$\tilde{∆}:={∆}_{X=0}=\frac{s_{px}\cdot d}{f_{m}\cdot d_{\mathrm{ML}}}.$$
We then calculate the pixel disparity based on Equation (Equation 6) as
(9)$$\begin{array}{c} \tilde{S}\left(o\right):=S_{X=0}\left(o\right)=\tilde{∆}\cdot \frac{o\left(f_{M}-d\right)+f_{M}\cdot d}{o\cdot f_{M}} \end{array}$$
along with the object distance, which can be simplified to
(10)$$\begin{array}{c} \tilde{o}\left(S\right):=o_{X=0}\left(S\right)=\frac{f_{M}\cdot d\cdot \tilde{∆}}{S\cdot f_{M}-\tilde{∆}\left(f_{M}-d\right)}. \end{array}$$ The relative error of the shift is then calculated by
(11)$$\begin{array}{c} ERR_{S}\left(o\right):=\frac{\tilde{S}\left(o\right)-S\left(o\right)}{S(o)}=X\cdot \frac{d\cdot f_{M}-o\left(d-f_{M}\right)}{o\cdot f_{M}\left(d-X\right)}. \end{array}$$
By describing o as a multiple of the focus distance $o_{f}$, i.e., $o=\lambda \cdot o_{f}$, we obtain
(12)$$\begin{array}{c} ERR_{S}\left(\lambda \right):=ERR_{S}\left(\lambda \cdot o_{f}\right)=\frac{X(\lambda -1)}{\lambda o_{f}\cdot \left(\frac{X}{d}-1\right)}. \end{array}$$
For the error of the object or refocusing distance, two cases are analyzed. First, it is assumed that the correct shift *S* corresponding to the ground truth o is given; in a second step, the oversimplified model of Equation (Equation 10) is used to calculate the object distance. This error can be found in applications measuring the correct shift, e.g., by repeatedly refocusing an image and subsequently using the incorrect object distance calculation in order to estimate the associated metric distances in the scene. This error can be formulated as
(13)$$\begin{array}{cc} \; & ERR_{\tilde{o}}\left(o\right):=\frac{\tilde{o}\left(S\left(o\right)\right)-o}{o}=-\frac{X\left(d-f_{M}\right)+\frac{o}{o_{f}}X\left(f_{M}-d\right)+Xf_{M}d\left(\frac{1}{o_{f}}-\frac{1}{o}\right)}{f_{M}\left(d-X\right)+\frac{o}{o_{f}}X\left(f_{M}-d\right)+Xf_{M}d\frac{1}{o_{f}}}. \end{array}$$
Using $o=\lambda \cdot o_{f}$ again, we obtain
(14)$$ERR_{\tilde{o}}\left(\lambda \right):=ERR_{\tilde{o}}\left(\lambda o_{f}\right)=\frac{X{\left(\lambda -1\right)}^{2}}{\lambda o_{f}\left(1-\frac{X}{d}\right)-X\left(\lambda -1\right)\lambda }.$$
The second case assumes an incorrectly calculated shift *S* based on Equation (Equation 9), which is subsequently used to refocus an image with a refocusing algorithm complying with the correct object distance estimation in Equation (Equation 7). This type of error is provided by
(15)$$\begin{array}{cc} \; & ERR_{\tilde{S}}\left(o\right):=\frac{o(\tilde{S}\left(o\right))-o}{o}=\frac{f_{M}∆-f_{M}X\tilde{∆}\left(\frac{1}{o}-\frac{1}{o_{f}}\right)}{\tilde{∆}\left(1-\frac{o}{o_{f}}\right)\left(f_{M}-X\right)+∆f_{M}\frac{o}{o_{f}}}-1. \end{array}$$
After substituting $o=\lambda \cdot o_{f}$, the error can be formulated by
(16)$$\begin{array}{c} ERR_{\tilde{S}}\left(\lambda \right):=ERR_{\tilde{S}}\left(\lambda o_{f}\right)=\frac{X{\left(\lambda -1\right)}^{2}}{\lambda o_{f}\left(\frac{X}{f_{M}}-1\right)-\lambda^{2}X}. \end{array}$$
Now, assuming a camera with a focal length of $f_{M}=100$ mm focused at a finite distance, Figure 5 shows exemplary error values for different values of *X* relative to the focal length.

The visualization shows that all errors diverge for λ→0 with a rate depending on the positional relationship between the exit pupil and the principal plane $H_{\mathrm{cam}}$. Beyond the focus distance at λ=1, the object errors again diverge, while the shift error converges according to $ERR_{S}\left(\lambda \right)\underset{\lambda \to \infty }{⟶}\frac{X}{o_{f}\left(X/d-1\right)}$. Note that these graphs present an ideal refocusing case free of aliasing artifacts and limiting optical properties such as the depth of field; hence, our later experiments only verify a section of these results within the respective physical and image processing limits.

In summary, these examples show a large deviation between the estimated refocusing distances in models with and without consideration of the exit pupil whenever there is a non-zero distance *X* between the exit pupil and $H_{\mathrm{cam}}$. This leads to the question of how prevalent a significant X≠0 is in off-the-shelf main lenses. To answer this question, the data of 866 DSLR lenses listed by Claff [47] were collected, then *X* and $f_{M}$ for each lens were calculated via paraxial ray tracing. The resulting data in Figure 6 show a nearly linear connection between the focal length of a lens and the distance *X*, with a Pearson correlation coefficient of 0.8994. Fitting a linear model to these data results in the non-zero function $X\left(f_{M}\right)=0.7108\cdot f_{M}-56.5546$ with coefficient of determination $R^{2}=0.8089$. Further examination shows that only a small subset, 62 of the 866 lenses, exhibits values for *X* below 5% of the focal length, i.e., $|X|<0.05f_{M}$. On the other hand, for the other 627 lenses the deviation is larger than $|X|>0.25f_{M}$, and 444 lenses even have values $|X|>0.5f_{M}$. Overall, these data show that the assumption of X≈0 is usually not met in reality. Therefore, the exit pupil should be considered when relating the camera-side light field to the scene’s light field.

## 4. Revisiting SPC Methods

In this section, several previous works are examined with respect to the exit pupil’s role in the respective model deductions.

### 4.1. Equivalent Ray Model

First, the equivalence between the refocusing model in Equation (Equation 7) and the ray intersection model presented by Hahne et al. [9] is proven. Instead of basing the model on the decoding scheme of Dansereau et al. [7] in [9], an approach building upon the intersection of chief rays is presented in order to calculate the refocusing distance for a resampling of the raw plenoptic camera image. A comprehensive notation transfer into our setup is provided in Section B.1.

As depicted in Figure 7, the basic idea of [9] is to select two pixels on the sensor that show scene points from the desired focus plane, then trace rays from these through the respective microlens centers. The resulting camera-side intersection determines the distance of the virtual image inside the camera from the main lens; accordingly, the thin lens equation can be applied in order to calculate the corresponding object or refocusing distance. Without loss of generality, the following calculations assume an MLA with one microlens center located on the optical axis of the main lens.

For a given a sub-aperture image shift *S* in a pixel, as depicted in Section 2.3, this translates to a metric pixel disparity $\hat{S}$ on the sensor by
(17)$$\hat{S}=-\frac{s_{px}}{S},$$
with the sign flip resulting from the difference in conventions between this work and [9]. Under the premise of a well-configured plenoptic camera with a regular microlens grid, any two pixels from neighboring microlenses with a disparity of $\hat{S}$ can be chosen to calculate the image distance. To simplify the calculations, the first ray is chosen to run along the optical axis, as shown in Figure 7, and the second ray is based on the pixel at sensor position $d_{\mathrm{MLI}}+\hat{S}$ passing through the neighboring microlens center. According to [9] (compare Section B.1) this leads to two ray functions
(18)$$f\left(z\right)=0\;\mathrm{and}\;\tilde{f}\left(z\right)=\frac{d_{\mathrm{ML}}-\left(d_{\mathrm{MLI}}+\hat{S}\right)}{f_{m}}\cdot z+d_{\mathrm{ML}}$$
which intersect at
(19)$$z_{i}=-\frac{d_{\mathrm{ML}}\cdot f_{m}}{d_{\mathrm{ML}}-\left(d_{\mathrm{MLI}}+\hat{S}\right)}.$$
The crucial element that sets [9] apart from the works reviewed in the following sections is the correct calculation of the microlens image center distance $d_{\mathrm{MLI}}$ based on the exit pupil (compare the calculation of $u_{c,j}$ in Table A5) via
(20)$$d_{\mathrm{MLI}}=\frac{d_{\mathrm{ML}}}{d-X}\cdot f_{m}+d_{\mathrm{ML}},$$
which yields
(21)$$\begin{array}{cc} z_{i} & =-\frac{d_{\mathrm{ML}}f_{m}}{d_{\mathrm{ML}}-\left(\frac{d_{\mathrm{ML}}}{d-X}\cdot f_{m}+d_{\mathrm{ML}}+\hat{S}\right)}\\ \; & =\frac{\left(d-X\right)d_{\mathrm{ML}}f_{m}}{\left(d-X\right)\hat{S}+f_{m}d_{\mathrm{ML}}}. \end{array}$$
This intersection results in the image distance $i=d-z_{i}$ measured from $H_{\mathrm{cam}}$, and can be used to calculate the object distance by the thin lens equation via
(22)$$\begin{array}{cc} o & ={\left(\frac{1}{f_{M}}-\frac{1}{i}\right)}^{-1}={\left(\frac{1}{f_{M}}-\frac{1}{d-z_{i}}\right)}^{-1}\\ \; & =\frac{f_{M}\cdot d-f_{M}\cdot z_{i}}{\left(d-f_{M}\right)-z_{i}}\\ \; & =\frac{f_{M}\cdot d\cdot \left(\left(d-X\right)\hat{S}+f_{m}d_{\mathrm{ML}}\right)-f_{M}\cdot \left(d-X\right)d_{\mathrm{ML}}f_{m}}{\left(d-f_{M}\right)\cdot \left(\left(d-X\right)\hat{S}+f_{m}d_{\mathrm{ML}}\right)-\left(d-X\right)d_{\mathrm{ML}}f_{m}}\\ \; & =\frac{f_{M}\cdot d\cdot \frac{d-X}{f_{m}d_{\mathrm{ML}}}\hat{S}+f_{M}X}{\left(d-f_{M}\right)\cdot \frac{d-X}{f_{m}d_{\mathrm{ML}}}\hat{S}-f_{M}+X}\\ \; & =\frac{f_{M}\cdot \left(d\cdot ∆-S\cdot X\right)}{S\cdot \left(f_{M}-X\right)-∆\left(f_{M}-d\right)}, \end{array}$$
where Equation (Equation 17) and the definition $∆:=\frac{{∆}_{UV}}{{∆}_{ST}}=\frac{s_{px}\left(d-X\right)}{f_{m}d_{\mathrm{ML}}}$ from Section 2 are used in the last step. This equation equals the previously deduced Equation (Equation 7), proving the equivalence of both models.

### 4.2. Light Field Decoding and SPC Calibration

This section revisits the popular decoding and calibration theme presented by Dansereau et al. [7]. In that work, the raw plenoptic camera image is first de-multiplexed into an integer-indexed two-plane parameterization L(i,j,k,l). These indices are then transformed into metric rays and propagated through the main lens. The combination of these steps yields an intrinsics matrix
(23)$$\left[\begin{array}{c} s\\ t\\ u\\ v\\ 1 \end{array}\right]=\left[\begin{array}{ccccc} H_{1,1} & 0 & H1,3 & 0 & H_{1,5}\\ 0 & H_{2,2} & 0 & H_{2,4} & H_{2,5}\\ H_{3,1} & 0 & H3,3 & 0 & H_{3,5}\\ 0 & H_{4,2} & 0 & H_{4,4} & H_{4,5}\\ 0 & 0 & 0 & 0 & 1 \end{array}\right]\cdot \left[\begin{array}{c} i\\ j\\ k\\ l\\ 1 \end{array}\right]$$
associating the integer indices directly with metric coordinates (s,t,u,v) for the scene-side light field. Note that these do not correspond to the equally named coordinates in Section 2 which describe the camera-side light field coordinates before propagating them through the main lens.

The relevant step with respect to the gap between the exit pupil and the principal plane in this process is the division of the integer indices by the respective spatial frequencies of the pixels and microlenses via the matrix $H_{abs}^{\theta }$. As explained in Section 2.1, the grid of MICs corresponds to the scaled grid of microlens centers. Accordingly, the sampling rate for the microlens plane has to be scaled down, or equivalently, the pixel sampling rate has to be scaled up by the inverse factor. Dansereau et al. [7] acknowledged this fact and chose the second option by introducing a scaling factor, which in our notation (compare Section B.2) corresponds to
(24)$$M_{\mathrm{proj}}={\left(1+\frac{f_{m}}{d}\right)}^{-1}.$$
This scaling, however, assumes a projection center at the main lens principal plane. Using the exit pupil instead, the correct rescaling is provided by
(25)$$M_{\mathrm{proj}}={\left(1+\frac{f_{m}}{d-X}\right)}^{-1},$$
as visualized in Figure 8.

Fortunately, due to the overall formulation of the intrinsics as an end-to-end ray transformation, this slight change is not relevant for the calibration results, as neither $f_{m}$ nor *d* is directly estimated. Instead, the factor $M_{\mathrm{proj}}$ contributes to the intrinsic variables $H_{1,1}$ to $H_{4,4}$, and repeating the deduction of *H* with the correct scaling leads to the same general form for the intrinsics matrix as in Equation (Equation 23).

Similar cases of general parameters compensating for the model inaccuracies have been presented by Monteiro et al. [13] and Zhang et al. [12]. Both made use of the same decoding process as [7] and built upon the idea of directly relating the camera-side and scene light field. Monteiro et al. [13] slightly reduced the intrinsics matrix shown above, and subsequently used it to create an equivalent array of cameras for the scene-side light field. Zhang et al. [12] followed a similar approach by first relating the de-multiplexed light field in the form of sub-aperture images to the scene-side light field metric, which they then based all of their further calculations on. In both of these works, the interpretation of the intrinsic matrix parameters is irrelevant and they are not directly used to reconstruct the main lens properties; accordingly, these methods do not require any reformulation using a more complex main lens model.

Despite this, it is important to point out the inaccuracy in [7], as several related works only make use of the proposed decoding process and assume that a two-plane parameterization is received with a plane distance of *d* instead of d−X, as shown by the examples in the following section.

### 4.3. Depth Reconstruction

One such case is presented by Pertuz et al. [10] and repeated in the follow-up work by Van Duong et al. [14]. In this work, a model relating the sub-aperture image shift to the object distance was deduced, which translates into our notation (compare Section B.3) as
(26)$$o\left(\rho \right)=o_{f}\left(\frac{1-a_{0}\cdot \rho }{1-a_{1}\cdot \rho }\right)$$
with system-dependent parameters
(27)$$a_{0}=\frac{f_{m}\cdot d_{\mathrm{ML}}}{s_{px}\cdot d}\;\mathrm{and}\;a_{1}=\frac{o_{f}\cdot a_{0}}{f_{M}}$$
and shift parameter
(28)$$\rho =\frac{s_{px}\cdot \left(F-F^{\prime }\right)}{f_{m}\cdot d_{\mathrm{ML}}}.$$
There are two problems in the deduction of this model. First, the shift parameter ρ is not correctly deduced under the premise of light field data decoded by the method of Dansereau et al. [7]; second, the exit pupil is ignored. In the following, a corrected version is presented, which additionally explains how these two problems nearly neutralize each other and lead to the same general model, albeit with different parameter interpretations.

First, the incorrect shift parameter ρ is examined. In [10], this parameter is described as the pixel disparity between neighboring sub-aperture images obtained via the decoding process from Dansereau et al. [7], and as such should correspond to our shift parameter $\tilde{S}$, which also ignores the exit pupil. However, due to different conventions, ρ is positive when focusing to a distance larger than the focus distance, whereas $\tilde{S}$ is negative in that case (compare Equation (Equation 5) for α<1). Accordingly, ρ should equal $-\tilde{S}$; however, transforming these parameters into a common notation leads to
(29)$$\begin{array}{cc} -\tilde{S}\left(o\right) & =\tilde{∆}\cdot \frac{o\left(f_{M}-d\right)+f_{M}\cdot d}{-o\cdot f_{M}}\\ \; & =\frac{s_{px}\cdot d}{f_{m}\cdot d_{\mathrm{ML}}}\cdot \frac{o\left(f_{M}-d\right)+f_{M}\cdot d}{-o\cdot f_{M}}\\ \; & \neq \frac{s_{px}\cdot d}{f_{m}\cdot d_{\mathrm{ML}}}\cdot \frac{o\left(f_{M}-d\right)+f_{M}\cdot d}{d(f_{M}-o)}\\ \; & =\frac{s_{px}\cdot \left(d-\frac{o\cdot f_{M}}{o-f_{M}}\right)}{f_{m}\cdot d_{\mathrm{ML}}}\\ \; & =\frac{s_{px}\cdot \left(F-F^{\prime }\right)}{f_{m}\cdot d_{\mathrm{ML}}}=\rho . \end{array}$$
The reason for this discrepancy is the implicit incorrect assumption in [10] that the grid of microlens image centers equals the grid of microlens centers, i.e., $d_{\mathrm{ML}}=d_{\mathrm{MLI}}$. While a light field in general could be reparametrized with this step size in the ST plane, this requires exact knowledge of the camera and MLA geometry, which is usually unknown and not considered in the decoding process of [7]. Using the correct shift parameter for light field data by following [7] instead and rearranging the corresponding Equation (Equation 10) results in
(30)$$\begin{array}{cc} \tilde{o}\left(\tilde{S}\right) & =\frac{f_{M}\cdot d\cdot \tilde{∆}}{\tilde{S}\cdot f_{M}-\tilde{∆}\left(f_{M}-d\right)}\\ \; & =o_{f}\left(\frac{\left(d-f_{M}\right)\cdot \tilde{∆}}{\tilde{S}\cdot f_{M}-\tilde{∆}\left(f_{M}-d\right)}\right)\\ \; & =o_{f}\left(\frac{1}{1+\left(\frac{f_{M}}{\tilde{∆}\left(d-f_{M}\right)}\right)\cdot \tilde{S}}\right)=o_{f}\left(\frac{1}{1-\left(\frac{o_{f}\cdot f_{m}\cdot d_{\mathrm{ML}}}{d^{2}\cdot s_{px}}\right)\cdot \left(-\tilde{S}\right)}\right). \end{array}$$
While this correction of Equation (Equation 26) continues to ignore the exit pupil, just as [10] does, it is considerably simpler than the original model, as only a single system-dependent parameter $\frac{o_{f}\cdot f_{m}\cdot d_{\mathrm{ML}}}{d^{2}\cdot s_{px}}$ is present instead of the two parameters $a_{0}$ and $a_{1}$ in Equation (Equation 26).

Finally, by introducing a non-zero distance *X* and thereby using o and *S* instead of $\tilde{o}$ and $\tilde{S}$, the full model can be deduced from Equation (Equation 7) as
(31)$$\begin{array}{cc} o(S) & =\frac{f_{M}\cdot \left(d\cdot ∆-S\cdot X\right)}{S\cdot \left(f_{M}-X\right)-∆\left(f_{M}-d\right)}\\ \; & =o_{f}\left(\frac{\left(d-f_{M}\right)\cdot \left(f_{M}\cdot d\cdot ∆-f_{M}\cdot X\cdot \left(-S\right)\right)}{\left(f_{M}\cdot d\right)\cdot \left(∆\left(d-f_{M}\right)-\left(X-f_{M}\right)\cdot S\right)}\right)\\ \; & =o_{f}\left(\frac{f_{M}\cdot d\cdot ∆-f_{M}\cdot X\cdot S}{f_{M}\cdot d\cdot \left(∆-\frac{X-f_{M}}{d-f_{M}}\cdot S\right)}\right)\\ \; & =o_{f}\left(\frac{1-\left(\frac{X}{∆\cdot d}\right)\cdot S}{1-\left(\frac{X-f_{M}}{∆\cdot (d-f_{M})}\right)\cdot S}\right). \end{array}$$
To align this model with the inverted shift direction of [10], we define
(32)$$\begin{array}{c} o_{f}\left(\frac{1-\left(\frac{X}{∆\cdot d}\right)\cdot S}{1-\left(\frac{X-f_{M}}{∆\cdot (d-f_{M})}\right)\cdot S}\right)=o_{f}\left(\frac{1-\left(\frac{-X}{∆\cdot d}\right)\cdot \left(-S\right)}{1-\left(\frac{f_{M}-X}{∆\cdot (d-f_{M})}\right)\cdot \left(-S\right)}\right)=:o_{f}\left(\frac{1-a_{0}\cdot \left(-S\right)}{1-a_{1}\cdot \left(-S\right)}\right). \end{array}$$
This model has the same general form as Equation (Equation 26) proposed by Pertuz et al. [10], which explains the reasonable experimental results in that work. Nevertheless, the interpretation of the system-dependent parameters $a_{0}$ and $a_{1}$ as provided in [10] and repeated in [14] is incorrect, which is verified by experiment (V) in the following section. This different interpretation could lead to problems when the model needs to be fitted to data and the initial parameter values are based on the incorrect direct calculation.

Overall, the results of Pertuz et al. [10] are not entirely incorrect; however, the light field representation based on [7] simply does not match the implicit assumptions used for the model deduction. More specifically, the main problem of [10] is the definition of the shift parameter ρ for light field data, which is decoded similarly to [7] but is based on known microlens centers instead of MICs. While the light field could theoretically be reparametrized using the parallel projections of the microlens centers onto the sensor, this would require exact knowledge of the SPC intrinsics, namely, the parameters of the MLA and its placement relative to the main lens and sensor.

## 5. Evaluation

### 5.1. Simulation Environment

Because SPCs with exchangeable lenses were not commercially available at the time of writing and custom-built solutions are costly as well as prone to misalignment of the optical components, we resorted to synthetic experiments via extension of the ray tracing solution we provided in [40]. Our publicly available [16], updated version of the Blender [11] add-on expands the original simulation in the following aspects:Simulation of aspherical lenses and zoom lenses.Configurable MLA pose, thickness, and IOR.Automatic focusing with lens group movement based on paraxial approximations.Integration of Claff’s lens collection [47] and a collection of sensor presets.Assisted plenoptic camera (SPC and FPC) configuration based on the ideas of [48].

This simulation setup facilitates quick generation of a broad range of plenoptic cameras, such as the example shown in Figure 9, and is used in the following to validate the formal analysis from Section 2.3, Section 3, and Section 4.3.

### 5.2. Experiments

For the validation, the five lenses listed in Table 1 were selected from the database [47]. While the first lens presents the ideal case of X≈0 mm, i.e., with the exit pupil coinciding with the camera-side principal plane, the remaining four lenses present interesting cases with varying relationships between *X* and the focal lengths $f_{M}$ of the lenses.

Each of these lenses was used in two SPC configurations: one with a finite focus distance $o_{f}<\infty$ and another focused at infinity $o_{f}=\infty$. To this end, the MLA placement with respect to the main lens, i.e., the distance *d* between the MLA and the camera-side principal plane of the main lens, was calculated using the thin lens Equation (Equation 1), as $d=\frac{f_{M}\cdot o_{f}}{o_{f}-f_{M}}$ for the finite case, and simply set to $d=f_{M}$ for $o_{f}=\infty$. The remaining microlens parameters were automatically initialized using the main lens and sensor properties so as to fulfill the following two constraints. First, the microlens f-number needs to match that of the main lens in order to optimally cover the sensor area [4]; second, a predefined number of 129×129 microlens images should be visible on the sensor in order to guarantee this resolution for the sub-aperture images. The resulting parameters were then fine-tuned by hand to accommodate the approximating nature of the f-number constraint and guarantee that the MICs coincided with the centers of the sensor pixels. This optimal MIC positioning has two effects. First, it renders the resampling during the decoding process of [7] unnecessary. After normalizing the raw images with white images to get rid of vignetting effects [50], the sub-aperture images in this setup can be directly extracted by combining the same relative sensor pixels from each microlens image [4]. Second, the evaluation can then concentrate on validating the refocusing itself instead of additionally dealing with compensating for interpolation artifacts from the decoding process. The full experimental setups can be found in Appendix D. Five experiments were performed with each of the ten following setups.

**(I) MICs and Exit Pupil:** The exit pupil as origin of the MICs is verified by first tracing ray bundles from the main lens aperture center through the main lens and MLA onto the sensor. This results in a set of sensor hits for every microlens. Due to the small variance within such a set, the mean is considered to represent the ground truth position of the microlens image center. In a second step, rays are traced from these sensor positions through the corresponding microlens centers and the convergence location of the resulting ray bundle is calculated in two ways: first by performing a line search along the optical axis for the minimum blur spot position of the ray bundle, and second based on the rays’ intersections with the optical axis. For these intersections, the mean and variance are calculated and presented alongside the minimum blur spot position. This whole process is visualized in Figure 10.

**(II) o and S:** A calibration pattern (more specifically, a Siemens star with four spokes) is placed at various distances in front of the camera. After demultiplexing the plenoptic camera image, the sub-aperture image shift *S* required to focus onto the given target distance o is measured. This is accomplished via line search, i.e., by repeatedly refocusing the image with a simple shift-and-sum algorithm [4] and calculating the sharpness of the refocused image. Here, the variance of the Laplacian [51] is used as the metric for image sharpness and the shift value with the highest score is considered the optimum. This procedure results in tuples of ground truth distances o and measured shifts *S*, which are then used to verify the connection S(o) as formally described in Equation (Equation 6).

To validate the inverted connection o(S) in Equation (Equation 7), all images are refocused for a given set of shift values. For each of these shift values, the object distance associated with the best focused image is considered the measured object distance for the respective shift value. The resulting tuples of preset shifts and measured distances are then used to verify o(S).

**(III) ${\mathrm{ERR}}_{\tilde{o}}$ Validation:** The data from experiment (II) are further used to verify the error model $ERR_{\tilde{o}}$. In detail, the measured shift *S* for a known target distance o is used along with Equation (Equation 10) to approximate $\tilde{o}\left(S\right)$ and calculate the measured relative error according to Equation (Equation 13). This error is then compared to the expected error obtained by directly calculating *S* based on the camera’s properties instead of measuring it.

**(IV) ${\mathrm{ERR}}_{\tilde{S}}$ Validation:** In addition, the images of (II) can be used to verify the error model $ERR_{\tilde{S}}$ presented in Section 3. First, for every target distance o, the incorrect shift $\tilde{S}\left(o\right)$ is calculated based on the assumption X=0 as in Equation (Equation 9), and the images of the patterns at different positions are all refocused with this parameter. The target distance corresponding to the sharpest of the refocused images approximates $o(\tilde{S})$, and is then used to measure $ERR_{\tilde{S}}$ as in Equation (Equation 15). Again, the measured values are compared to the errors obtained by directly calculating $o(\tilde{S})$.

Note that instead of verifying $ERR_{\tilde{o}}$ and $ERR_{\tilde{S}}$, Equations (Equation 9) and (Equation 10) and the shift error model in Equation (Equation 11) could be directly validated with the data measured and calculated in experiments (III) and (IV). However, these equations do not include a comparison between the incorrect estimations and the ground truth refocusing distances. Therefore, the indirect validation of those models by means of the resulting refocusing errors was preferred.

**(V) Validation of Section 4.3:** As analyzed in that section, the overall formulation of the model presented by Pertuz et al. [10] is correct; however, the model parameters have a different interpretation under the assumption of light field data decoded following the method from [7]. To verify the corrected model, the parameters $a_{0}$ and $a_{1}$ were first calculated based on the formula of Pertuz [10], then according to our model from Section 4.3, and finally fitted to the data of (II), i.e., the set of shift–distance pairs with measured shifts and ground truth target distances. For this parameter fitting, a grid search for the best parameters by means of the RMSE was performed with a grid explicitly containing both directly calculated parameter sets.

### 5.3. Results and Discussion

**(I)** As shown in Figure 11, the ray bundle consisting of rays running from the calculated ground truth MICs through the microlens centers generally converge towards the exit pupil in all setups. The closer the ray origins are to the optical axis (for example, compare the inner 25% of rays shown in Figure 11), the closer the minimum blur spot is located to the exit pupil. The deviations for larger sets of rays including the outermost MICs, as especially visible for the Canon setups, can be explained by the dependence of the exit pupil’s location on the viewing angle. Similar to the curved focal plane in lenses with a significant non-zero Petzval field curvature, the exit pupil cannot be well approximated by a plane in certain setups, and consequently affects the MIC grid on the sensor in a nonlinear fashion [42]. This is a clear limitation of our model, which is built upon paraxial approximations. Nevertheless, considering that the origin of the MICs is located close to the exit pupil plane even in extreme cases such as the outermost microlenses in the Canon setups, these results again confirm the observation of Hahne et al. [8], and justify the recommendation to examine the necessity of including the exit pupil in the lens model.

**(II)** The connection between the focus distance o and the shift *S* as described by Equation (Equation 6) is verified by the measurements presented in Figure 12. The mean of the absolute differences ${|S\left(o\right)}_{measured}-S{\left(o\right)}_{GT}|$ between the measured and directly calculated ground truth shift values across all ten setups is 0.008 px, with a variance of 0.0026 px. The worst single setup is the finitely focused Olympus setup, with a mean of absolute differences of 0.04 px and a variance of 0.025 px. These values are in the range of expected inaccuracies resulting from the image processing methods involved; in particular, the interpolation steps required by the shift-and-sum refocusing [4], as well as the rather simple (de)focus measure, which is prone to interference errors, are limiting factors that prevent higher accuracy.

A similar situation is observable in the inverse case, i.e., for the model o(S) from Equation (Equation 7), as presented in Figure 13. Due to the wide range of target distances, in this case the relative absolute differences $\frac{{|o\left(S\right)}_{measured}-o{\left(S\right)}_{GT}|}{o{\left(S\right)}_{GT}}$ are calculated to quantify the results from Figure 13. For the finite setups, the mean of these relative absolute differences is 0.13%, with a variance of $2\cdot 10^{-5}$%, while in the infinite cases the overall mean is 0.72% with a variance of 0.01%. Here, the infinitely focused Zeiss setup has the worst performance, with a relative absolute difference mean of 0.9% and a variance of 0.004%. The worse performance of the infinitely focused setups is a consequence of the smaller range of shift values representing a larger refocusing range (compare Figure 13), resulting in greater susceptibility to small shift changes. Nevertheless, the overall performance again confirms the model o(S) within the constraints posed by the involved image processing steps.

**(III)** The results of the $ERR_{\tilde{o}}$ verification experiment are shown in Figure 14. The graphs again show the error based on measurements compared to the directly calculated ground truth error based on the error models presented in Section 3.

The mean of the absolute differences between the measured and calculated error values is 0.003, with a variance of $10^{-5}$, which validates our models within the limitations of only the accuracy of the image processing methods and the optical properties of the chosen lenses. More specifically, the small deviations from the expected values, which are present even in the baseline case for the Rodenstock lens, can be explained as due to the interpolation operations required in Experiment (II) by the shift-and-sum refocusing algorithm [4] for non-integer shift values. The larger deviation visible at close range for the Olympus lens with finite focus distance is a result of the optical limits of this lens, which can be explained as follows. A single sub-aperture image contains one pixel per microlens; hence, such an image can be considered sharp if the calibration target is imaged onto the MLA with all blur spot sizes being smaller than the microlens diameter $d_{\mathrm{ML}}$. For the Olympus lens, however, these blur spot sizes increase more drastically at close range than those of the other lenses, leading to severe defocus blur even in the sub-aperture images. This can produce interference artifacts during the refocusing, leading in turn to fluctuating contrast measurements. Because the refocus distance o is determined by these measurements, this affects the error calculated by $\frac{\tilde{o}\left(S\left(o\right)\right)-o}{o}$ in Equation (Equation 13).

**(IV)** Experiment (III) validated the error model in the case of a correctly estimated shift combined with an oversimplified distance estimator. Figure 15 shows that the inverse problem of an incorrectly calculated shift with the shift-and-sum refocusing is modeled correctly as well. The mean of absolute differences between the measured and directly calculated error values is 0.004, with a variance of $2.47\cdot 10^{-5}$.

Overall, Experiments (III) and (IV) confirm the formal deductions in Section 3, and again justify our warning to mind the exit pupil when modeling a standard plenoptic camera. However, the results also hint at further minor optical or algorithmic aspects not being accounted for. In all cases, the mean of the absolute differences between the measured values and the ground truth model is up to two orders of magnitude larger than the respective variance. This is the result of nearly constant over- or underestimation, and could indicate a systematic error, which could be caused by, e.g., the refocusing model being limited to paraxial calculations.

**(V)** Regarding the correction of the model from [10] in Section 4.3, the results presented in Table 2 show that the corrected model appropriately describes the connection between the refocusing distance and the sub-aperture image shift. In all setups, the RMSE of our directly calculated model was within 1.75 mm of the RMSE of the fitted model. In addition, the fitted parameters are approximated well by the direct calculation with our model. On the other hand, the model in [10] can be regarded as incorrect for this light field parameterization, with one exception: the parameter $a_{1}$ in the setups focused at infinity is in general correct, which can be explained as follows. Neither the original model of [10] nor our correction directly considers the case $o_{f}=\infty$; instead, a large focus distance of $o_{f}=10^{6}\;m$ is used as an approximation in both cases. For such a focus distance, the distance *d* between the main lens and the MLA is close to the focal distance of the main lens, i.e., $d=f_{M}+\epsilon$ for some small value ϵ>0. With this formulation, the correctness of $a_{1}$ in the infinite cases can formally be explained by
(33)$$\begin{array}{cc} \frac{a_{1,\mathrm{Pertuz}}}{a_{1,\mathrm{our}}} & =\left(\frac{o_{f}\cdot a_{0,\mathrm{Pertuz}}}{f_{M}}\right)\cdot \left(\frac{∆(d-f_{M})}{f_{M}-X}\right)=\frac{o_{f}\cdot \frac{f_{m}\cdot d_{\mathrm{ML}}}{s_{px}\cdot d}}{f_{M}}\cdot \frac{\frac{s_{px}\cdot \left(d-X\right)}{f_{m}\cdot d_{\mathrm{ML}}}\cdot \left(d-f_{M}\right)}{f_{M}-X}\\ \; & =o_{f}\cdot \underset{=1/o_{f}}{\underset{︸}{\frac{d-f_{M}}{df_{M}}}}\cdot \frac{d-X}{f_{M}-X}=\frac{f_{M}+\epsilon -X}{f_{M}-X}=1+\frac{\epsilon }{f_{M}-X}\approx 1. \end{array}$$

## 6. Conclusion and Limitations

Overall, this work shows that the exit pupil can play a crucial role when modeling the relation between the camera-side and scene-side light fields. The connection between sub-aperture image shift and refocusing distance is derived analogously to previous work but with the additional consideration of an exit pupil that does not coincide with the principal plane of the main lens. Based on this deduction, two error models for the relative refocusing distance are created and validated. A subsequent review of previous work shows that a sufficiently general formulation of the SPC calibration model in most methods absorbs these errors, albeit leading to an incorrect interpretation of the model parameters. As an example, a correction of the work of Pertuz et al. [10] is presented and validated.

Nevertheless, despite the good evaluation results, there are several limitations to this work. First of all, the experiments were performed on simulated data. While the ray tracing-based lens simulation has been verified to exhibit the optical properties stated in the respective lens patents, i.e., aberration and distortion measurement results from the patents could be reproduced, there still is a gap between simulation and reality. On one hand, the specified lens parameters could differ from the final production lenses due to manufacturing inaccuracies or even deliberate parameter obfuscation by the lens manufacturer to hide specific lens details. On the other hand, the used framework provided by Blender [11] does not include wave optic effects such as diffraction [52,53]. Without these effects, the simulated optics are not diffraction-limited, and might produce images that are sharper than their real pendants.

Further limitations concern the formal lens model used for the error deduction. First, the microlenses in our model are still formally regarded as thin lenses. While Hahne et al. [9] used a thick lens model with explicit microlens principal planes, it was decided to leave this aspect out of our theoretical discussion for reasons of clarity and comprehensibility. However, the microlens thickness was factored in while performing the experiments, and Appendix C shows that an extended model does not change the equations deduced in Section 2 and Section 3.

Another limitation, which does affect the validity of these equations, is the restriction to paraxial models, more specifically the repeated use of the thin lens equation Equation 1 in various calculations as well as fixed positions for the principal planes and exit pupils. The thin lens equation describes the relation between the object distance, the image distance, and the focal length of the lens, and is usually only valid along the optical axis. As the distance from this axis grows, third-order aberrations such as Petzval field curvature, i.e., a curved focus surface, can affect the refocusing distance [42]. Furthermore, as seen for the Canon lens evaluated in Experiment (I), the position of the exit pupil might vary depending on the viewing angle, leading to a reduction in the applicability of the deduced models.

Further work on the listed limitations is not expected to significantly alter the presented results, as these are already within the expected accuracy bounds set by the involved image processing steps. Instead, future work could target the second type of plenoptic cameras, namely, FPCs, for which a multitude of different calibration methods exists, and which also differ in their assumed lens models and could benefit from minding the gap between the principal plane and exit pupil.

## Figures and Tables

**Figure 2 sensors-24-02522-f002:**
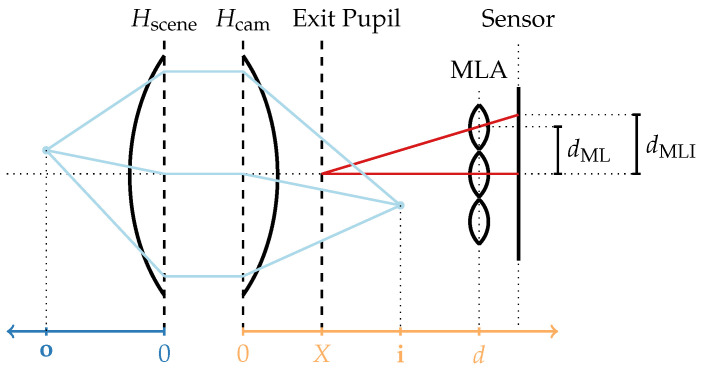
Plenoptic camera modeled by a thick main lens combined with a thin-lens MLA. The pitch of the microlens is described by $d_{\mathrm{ML}}$, the distance between the neighboring microlens image centers (MICs) is denoted as $d_{\mathrm{MLI}}$, *X* describes the distance between the exit pupil and the camera-side principal plane, and *d* is the distance between $H_{\mathrm{cam}}$ and the MLA. A complete notation overview is provided in Appendix A.

**Figure 3 sensors-24-02522-f003:**
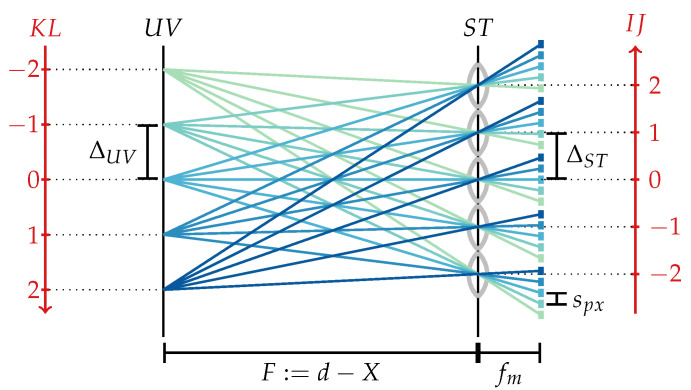
Integer (red) and metric (black) two-plane parameterization of the light field. Here, $s_{px}$ describes the size of a sensor pixel and $f_{m}$ the focal length of a microlens, which for an SPC coincides with the distance between the MLA and sensor.

**Figure 4 sensors-24-02522-f004:**
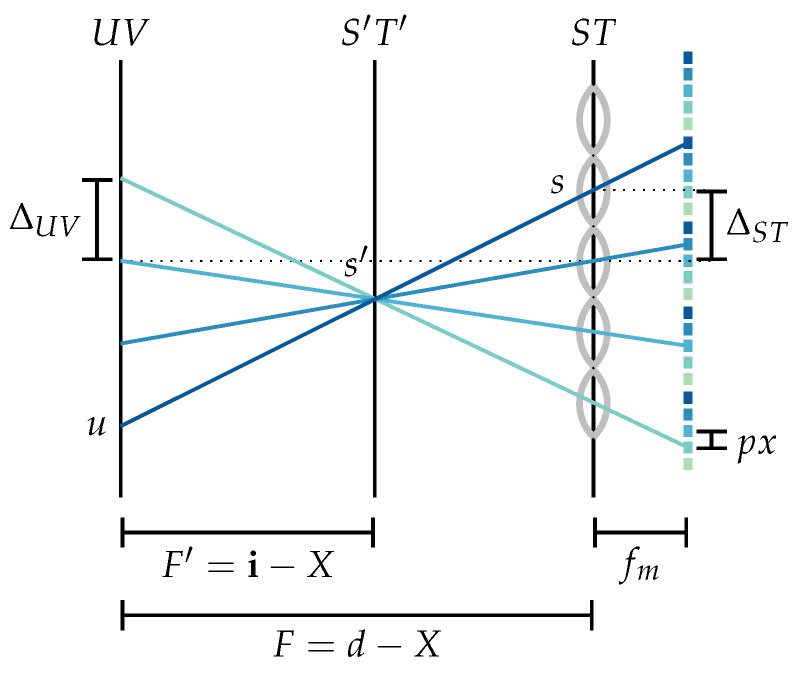
Light field refocusing via shifting of the virtual sensor, i.e., the ST-plane is moved to the image distance i. A ray $\left(s^{\prime },u\right)$ can be associated with a ray (s,u) by means of the triangle equality, i.e., $s=u+\frac{F}{F^{\prime }}\left(s^{\prime }-u\right)$.

**Figure 5 sensors-24-02522-f005:**
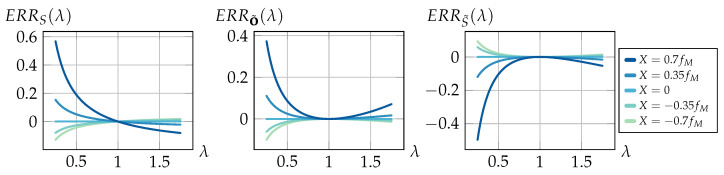
Left: Relative shift error based on $\lambda =\frac{o}{o_{f}}$. Middle/Right: The two cases of relative object distance errors for an assumed camera focused at a finite distance which is met for a relative distance of $\frac{o}{o_{f}}=1$. Negative error values indicate an underestimation of the ground truth value, while positive errors represent an overestimation.

**Figure 6 sensors-24-02522-f006:**
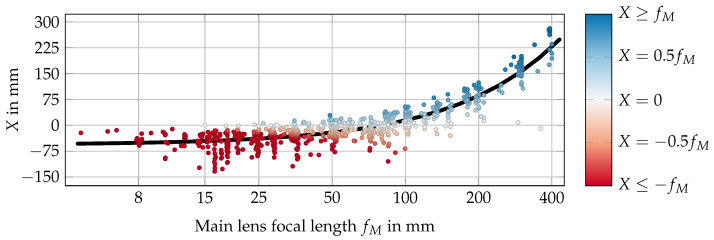
Distances *X* between the exit pupil and principal plane $H_{\mathrm{cam}}$ for 866 lenses [47] sorted by focal length. The black line represents the linear model fitted to the data, while the colors of the datapoints indicate the relationship $X/f_{M}$. Note that the horizontal axis uses logarithmic scaling due to the large number of lenses with a focal length below 100 mm.

**Figure 7 sensors-24-02522-f007:**
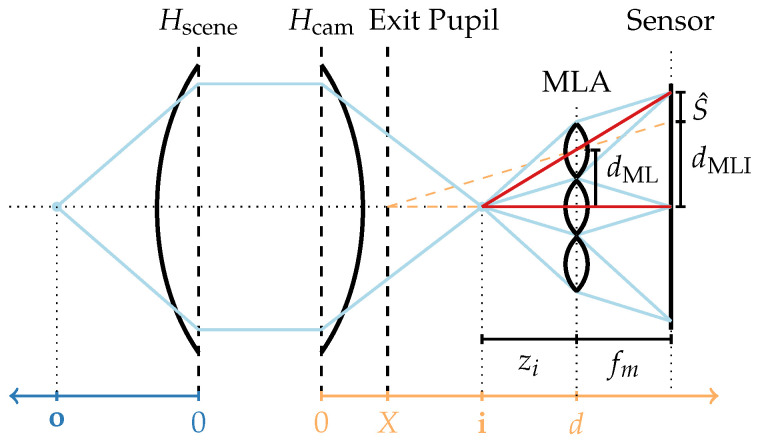
Image formation (light blue) for an object point located at distance o from $H_{\mathrm{scene}}$. The image of this point, located at distance i from $H_{\mathrm{cam}}$, is seen by multiple microlenses, and its projections onto the sensor have a metric disparity of $\hat{S}$. In order to determine the object distance o, Hahne et al. [9] proposed using intersecting ray functions (red) from two of the images and transferring the resulting image distance i to the scene via the thin lens equation.

**Figure 8 sensors-24-02522-f008:**
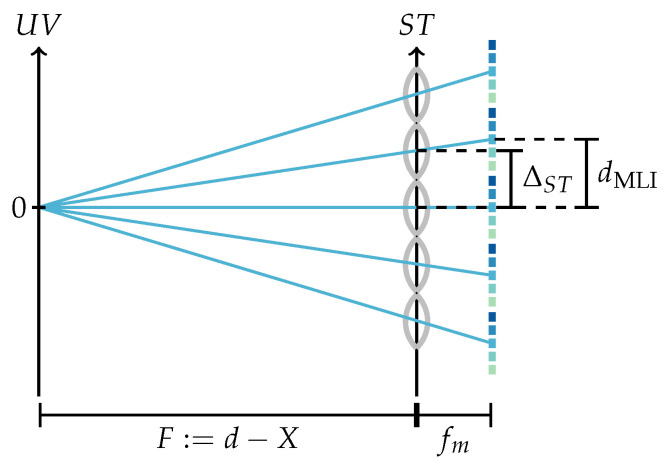
The central sub-aperture image consists of the MICs, i.e., the image of the aperture center viewed across all microlenses. These MICs originate from the center of the UV plane, i.e., the exit pupil; accordingly, the distance between neighboring MICs is provided by the triangle equality via $d_{\mathrm{MLI}}={∆}_{ST}\cdot \left(f_{m}+F\right)/F={∆}_{ST}\cdot \left(1+f_{m}/\left(d-X\right)\right)$.

**Figure 9 sensors-24-02522-f009:**
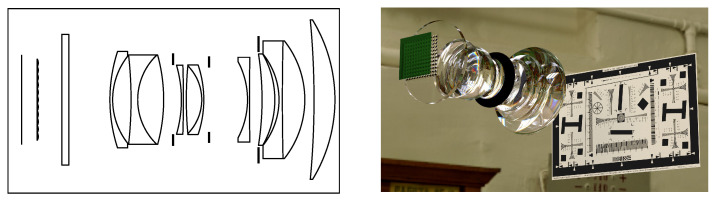
Cross-section and rendering of an example evaluation setup. A fully modeled lens was combined with a two-plane MLA model and a sensor in order simulate a plenoptic camera via ray tracing. Calibration patterns were placed at different distances in front of this setup to verify the analytical models. Note that the housing of the camera and lens were removed for the purpose of visualization. The ISO 12233 pattern [49] is used with the permission of Cornell University.

**Figure 10 sensors-24-02522-f010:**
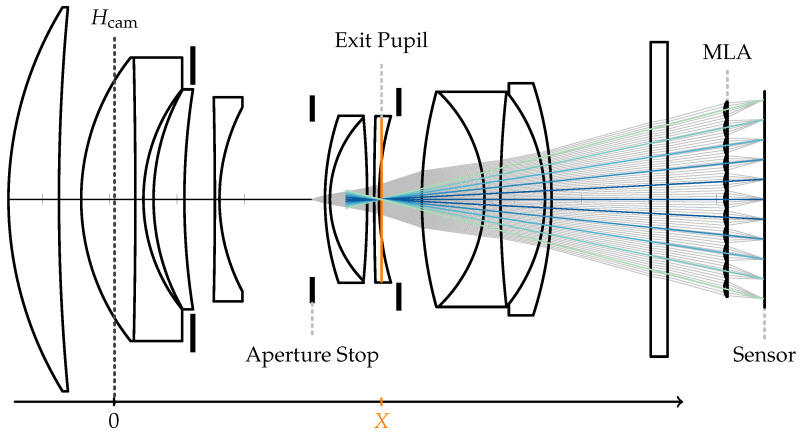
Experiment (I): The two steps of the MIC/exit pupil verification visualized for the Zeiss Batis 85mm F1.8. Rays (light gray) are traced from the main lens aperture center through the main lens and MLA onto the sensor. The resulting means of the sensor hits per microlens represent the MICs. The exit pupil as the approximate source of these points is verified by backwards tracing of the rays (blue/green) from the MICs through the respective microlenses and calculation of the minimum blur spot position along with the mean and variance of intersections along the optical axis.

**Figure 11 sensors-24-02522-f011:**
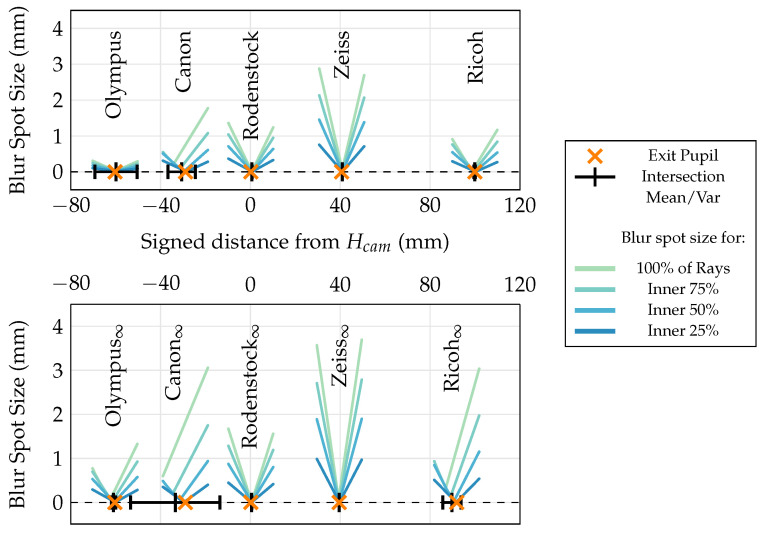
Results of Experiment (I). The orange markers indicate the locations of the exit pupils on the optical axis (horizontal, dotted line) with respect to the respective principal planes ($H_{\mathrm{cam}}$). The black markers show the mean and variance of the intersection points between the optical axis and the rays traced back from the MICs through the microlens centers. The colored functions show the blur spot sizes for different subsets of these ray bundles close to the exit pupil (compare Figure 10). These subsets contain the respective portion of rays from the bundle which are closest to the optical axis.

**Figure 12 sensors-24-02522-f012:**
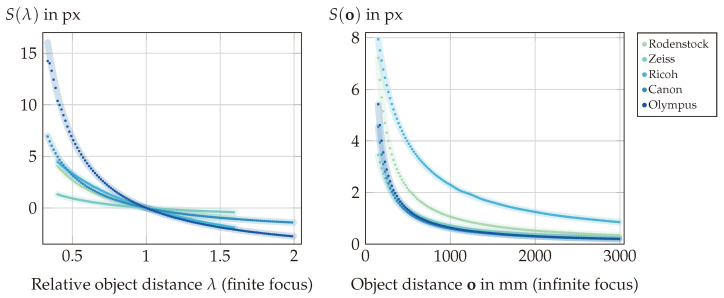
Results of Experiment (II): Verification of the model S(o) according to Equation (Equation 6). The datapoints represent measured shift values for the respective object distances, while the underlying lines represent the expected directly calculated values. Left: The error for the setups with finite focus based on the relative object distance $\frac{o}{o_{f}}$. Right: The error for setups focused at infinity.

**Figure 13 sensors-24-02522-f013:**
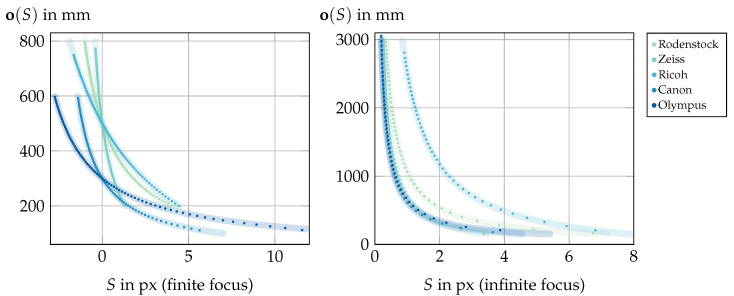
Results of Experiment (II): Verification of the model o(S) according to Equation (Equation 7). The datapoints represent measured focus distance values for the respective shifts and the underlying lines represent the expected directly calculated values.

**Figure 14 sensors-24-02522-f014:**
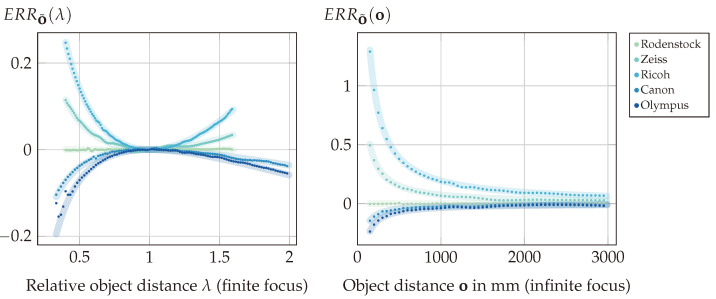
Results of Experiment (III): Relative object distance error resulting from a correctly estimated shift with incorrect object distance estimation based on the assumption that X=0. The thick lines indicate the predicted ground truth error based on Equations (Equation 13) and (Equation 14), with the points indicating the measurements. Left: The error for the setups with finite focus. To provide a comparative visualization of the different target distance ranges, the results are shown based on the relative object distance $\lambda =\frac{o}{o_{f}}$. Right: The error for the setups focused at infinity.

**Figure 15 sensors-24-02522-f015:**
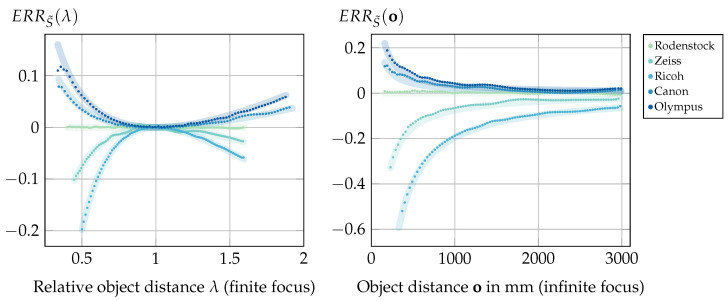
Results of Experiment (IV): Relative object distance error resulting from an incorrectly calculated shift ignoring the exit pupil in combination with a correct object distance estimation. The thick lines indicate the predicted ground truth error based on Equations (Equation 15) and (Equation 16), with the points indicating the measurements. Left: The error for the setups with finite focus. To provide a comparative visualization, the results are shown based on the relative object distance $\frac{o}{o_{f}}$, i.e., the finite focus distance for the respective setups is met at $\frac{o}{o_{f}}=1$. Right: The error for the setups focused at infinity.

**Table 1 sensors-24-02522-t001:** Overview of the simulated main lenses and their properties in the finite and infinite focus setup. Note that the focal length of a lens can vary in different setups due to the lens group movements involved in refocus or zoom operations.

	Finite Focus				Infinite Focus		
Lens Model	$f_{M}$ (mm)	X (mm)	$\frac{X}{f_{M}}$	Focus Dist. (mm)	$f_{M}$ (mm)	X (mm)	$\frac{X}{f_{M}}$
Rodenstock Sironar-N 100 mm F5.6	99.998	0.194	0.002	500.0	99.998	0.194	0.002
Zeiss Batis 85 mm F1.8	82.047	40.652	0.496	500.0	82.860	39.573	0.478
Ricoh smc Pentax-A 200 mm F4 Macro ED	167.994	99.908	0.595	500.0	173.115	91.854	0.531
Canon EF 85 mm F1.8 USM	84.998	−28.938	−0.341	300.0	84.998	−28.938	−0.341
Olympus Zuiko Auto-Zoom 85–250 mm F5	85.120	−60.219	−0.708	300.0	85.004	−60.308	−0.709

**Table 2 sensors-24-02522-t002:** Results of Experiment (V): Model parameters and the resulting RMSE in mm for the ten SPC setups. The parameters for our method and that of Pertuz et al. from [10] were directly calculated based on the known optical properties. The fitted parameter sets were acquired via grid search and subsequent local optimization to fit the model to the data from Experiment (II).

	Pertuz et al. [10]			Ours (Equation (Equation 32))			Fitted		
Setup	$a_{0}$	$a_{1}$	RMSE	$a_{0}$	$a_{1}$	RMSE	$a_{0}$	$a_{1}$	RMSE
Rodens.	0.0927	0.4633	21.03	−0.00014	0.3705	0.64	−0.00038	0.36951	0.49
Zeiss	0.1853	1.1294	28.05	−0.13102	0.81304	1.51	−0.13109	0.81361	1.51
Ricoh	0.1140	0.3391	89.77	−0.07435	0.15079	0.52	−0.07403	0.15111	0.51
Canon	0.1341	0.4733	45.50	0.02630	0.36553	0.74	0.025547	0.36568	0.59
Olympus	0.0674	0.2376	37.49	0.02267	0.19286	1.12	0.0213	0.19146	0.62
Rodens._∞_	0.0927	55.588	96.87	−0.00018	55.495	7.19	0.00198	55.625	6.83
Zeiss_∞_	0.1070	322.73	153.30	−0.09773	322.53	9.67	−0.08574	323.61	8.35
Ricoh_∞_	0.0635	366.78	364.78	−0.07175	366.65	6.76	−0.06887	367.54	5.03
Canon_∞_	0.1458	1715.1	67.40	0.03702	1715	10.26	0.04576	1724.1	8.69
Olympus_∞_	0.1370	96.683	46.52	0.05680	96.603	8.23	0.06014	96.631	8.07

## Data Availability

Data are available in two publicly accessible repositories The data presented in this study are openly available at https://gitlab.com/ungetym/SPC-revisited, accessed on 22 February 2024, and the camera simulation is available at https://gitlab.com/ungetym/blender-camera-generator, accessed on 22 February 2024.

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
