# Peer review of "Mind the Exit Pupil Gap: Revisiting the Intrinsics of a Standard Plenoptic Camera"

_sensors, 2024, doi:10.3390/s24082522_

Round 1

Reviewer 1 Report

Comments and Suggestions for Authors

In this manuscript, the authors address the phenomenon of overlooking the exit pupil in the calibration of light field cameras, and derive a refocusing model considering the exit pupil.  Mathematical analysis and approach look fine. My concerns are as follows:

1. Page 5, what is the meaning of spx in the formula?

2. In the analysis of errors of Figure 5, what do positive and negative values represent? Do negative and positive errors indicate whether the focusing distance is before or after the exit pupil position?

3. Page 10, a model relating sub-aperture image shift to the object distance is deduced from Reference 13. In this paper, one contribution is a formal deduction of the connection between object distance and sub-aperture image shift. The deduction has been deduced by predecessors, so what is the significance of this work?

4. What simulation software is used in the experiment?

5. English spell check is required, and please recheck the format of references.

Comments on the Quality of English Language

English spell check is required. The authors should improve their language in the manuscript.

Reviewer 2 Report

Comments and Suggestions for Authors

---------------------
Summary:
---------------------

This paper proposes a model for standard plenoptic camera explicitly including the exit pupil.
The authors show that oversimplified main lens model leads to an incorrect interpretation of metric model parameters for most lenses used in the experiments.

They highlight the importance of the exit pupil in the process of estimating metric (scene side) quantities through refocusing shift, and therefore through the decoding process.
Even if realistic parameters do not matter in the calibration process, it usually generates wrong values for the parametrization and thus wrong metric estimation.
They derived two error models and provided an analysis of the role of the exit pupil in methods from the literature.
They verified through a ray-tracing based simulation of various standard plenoptic cameras using real lens data from datasheet.

---------------------
General comments:
---------------------
The article is overall easily understandable, well written and clear.
The mathematical derivations are sound.
The appendices are well appreciated, especially the summary of notations.
Source code and data used in the paper have been made publicly available.

The various experiments in simulation seem to prove the validity of the proposed model despite the lack of evaluation on real data.
The paper is overall good and well-motivated however I still have some concerns.

---------------------
Major remarks:
---------------------

1) The main issue lies in the lack of experiments using real-world data.
The model has been validated in simulation only, neglecting issues that might rise in real-world optics (manufacturing error, optics misalignments, aberrations, distortions, or diffraction effect).
The paper would have strengthen by validating at least on standard plenoptic camera available such as the Lytro Illum.
Furthermore, the authors indicated that other paper such as Hahne et al. [8,10] also considered the exit pupil in their process, but no evaluation or more detailed comparison is provided.

2) My other concern is that the authors do not provide a methodology to retrieve the overall parameters through calibration (especially including the position of the exit pupil) on real-world data.
Even if the proposed model proves to improve the estimation of metric parameters through the decoding process, it is not clear if based on real-world data we would be able to retrieve these intrinsic parameters.
Since generally calibration models are sufficiently general to permit the simplicity of a main lens model without considering the exit pupil, is a decoding method able to identify individually the X parameter?

---------------------
Minor remarks:
---------------------

- I would avoid terms such as "former" and "latter" that might be confusing for non-native English speakers. I would be more clear to directly refer to the mentioned methods. (e.g., l.144, l.148)
- l. 477, I think there is an error in the reference indicated, shouldn't it be [9]?
- no recent work is cited (except author's own work)

Furthermore, even though the authors mentioned the extension of their work to the focused/multi-focal plenoptic camera, they thus restricted themselves to the standard plenoptic camera here.
I think focused plenoptic camera (FPC) literature cannot be completely ruled out.
For instance:
- l.78, it is not clear why with SPC the problems arising from an ignored exit pupil are more directly observable in this case than with FPC? The authors need to elaborate more on this with references.
- MICs detection has been addressed also by Thomason et al. 2014; Noury et al. 2017; Suliga et al. 2018
- l.105, some missing methods from the literature provides more complex geometric model for the focused plenoptic camera that proved to be applicable to standard plenoptic camera as well (Labussiere et al. 2022)
- l.250, some methods from the FPC literature also take into consideration the scale between the MLA and the micro-images (Nousias et al. 2017; Noury et al. 2017; Labussiere et al. 2022)

---------------------
References:
---------------------

Thomason, C. M., Thurow, B. S., Fahringer, T. W. (2014). Calibration of a microlens array for a plenoptic camera. In: 52nd Aerospace Sciences Meeting, pp. 1–18.
Noury, C. A., Teulière, C., Dhome, M. (2017). Light-field camera calibration from raw images DICTA. In: 2017—International Conference on Digital Image Computing: Techniques and Applications, pp. 1–8
Suliga, P., Wrona, T. (2018). IEEE microlens array calibration method for a light field camera. In: Proceedings of the 19th International Carpathian Control Conference (ICCC), pp. 19–22
Nousias, S., Chadebecq, F., Pichat, J., Keane, P., Ourselin, S., Bergeles, C. (2017). Corner-based geometric calibration of multi-focus plenoptic cameras. Proceedings of the IEEE International Conference on Computer Vision, pp. 957–965.
Labussière, M., Teulière, C., Bernardin, F., Ait-Aider O. (2022). Leveraging Blur Information for Plenoptic Camera Calibration. In: Int J Comput Vis 130, pp. 1655–1677

Round 2

Reviewer 1 Report

Comments and Suggestions for Authors

This manuscript has been revised based on review's comment. I think that this paper could be accepted, if the author improves their language quality of the paper.

Comments on the Quality of English Language

The quality of English language is better than before.